# Should We Use Unilateral or Bilateral Tasks to Assess Maximal and Explosive Knee Extensor Strength in Patients with Knee Osteoarthritis? A Cross-Sectional Study

**DOI:** 10.3390/jcm10194353

**Published:** 2021-09-24

**Authors:** Jonas Pfeifle, David Hasler, Nicola A. Maffiuletti

**Affiliations:** 1Institut für Physiotherapie, Zürcher Hochschule für Angewandte Wissenschaften (ZHAW), Technikumstrasse 71, 8400 Winterthur, Switzerland; jonas.pfeifle@googlemail.com; 2Human Performance Lab, Schulthess Clinic, Lengghalde 2, 8008 Zurich, Switzerland; david.hasler@hest.ethz.ch; 3Institute for Human Movement Sciences and Sport, ETH Zurich, Winterthurerstrasse 190, 8057 Zurich, Switzerland

**Keywords:** inter-limb asymmetry, maximal voluntary strength, rate of force development, knee osteoarthritis, performance-based test, self-reported function

## Abstract

Deficits in maximal and explosive knee extensor strength, which are usually assessed with unilateral tasks, are substantial in patients with knee osteoarthritis (KOA). The aim of this study was to investigate the clinical relevance of unilateral vs. bilateral tasks for assessing knee extensor strength in patients with KOA. This was achieved primarily by comparing unilateral and bilateral inter-limb strength asymmetries and secondarily by examining the relationship between unilaterally and bilaterally measured strength, and performance-based and self-reported function. Twenty-four patients with unilateral KOA (mean age: 65 ± 7 years) performed isometric gradual and explosive maximal voluntary contractions to assess, respectively their maximal and explosive strength. Performance-based and self-reported function were also evaluated with standard functional tests and questionnaires, respectively. Inter-limb asymmetries of maximal and explosive strength did not differ significantly between unilateral (mean asymmetry: 26 ± 15%) and bilateral tasks (22 ± 21%). In the same way, the relationships between knee extensor strength—measured either unilaterally or bilaterally—and performance-based or self-reported function were not influenced by the type of task. In conclusion, it does not seem to make a difference in terms of clinical relevance whether maximal and explosive knee extensor strength are evaluated with unilateral or bilateral tasks in KOA patients.

## 1. Introduction

Knee osteoarthritis (KOA) is the most frequent musculoskeletal condition in older individuals [1]. KOA has a global prevalence of 23% among individuals aged > 40 years (~654 million individuals had KOA worldwide in 2020) and a pooled global incidence of 203 per 10,000 person-years in individuals aged > 20 years [2]. Patients with KOA are characterized by low levels of physical functioning in comparison to age- and sex-matched controls [3] due to a combination of joint pain, stiffness, and lower limb muscle weakness [3,4]. In particular, knee extensor strength is universally considered to be an important determinant of physical function in subjects with KOA [5].

Deficits in maximal and explosive knee extensor strength are considerable in patients with KOA [6,7,8]. Knee extensor strength is often evaluated using unilateral isometric maximal voluntary contractions (MVC) in both laboratory and clinical settings. These contractions can be either gradual, with the goal to attain the maximal force-generating capacity (often characterized by MVC torque), or explosive, with special emphasis on the rate of torque development (RTD). This latter outcome is increasingly considered due to its functional relevance with respect to daily living activities [9] and as it has been shown to be an independent predictor of physical performance in a large cohort study [10]. As such, unilaterally measured MVC torque and/or RTD are usually compared between the involved and uninvolved side of KOA patients, generally in the form of an inter-limb percent asymmetry, to provide an estimate of knee extensor muscle weakness [11]. Using this methodology, strength asymmetries comprised between 10% and 20% are arbitrarily referred to as “probably pathological”, while asymmetries greater than 20% are commonly considered as “pathological” (i.e., serious muscle weakness) [12].

Unilaterally measured knee extensor strength is closely related to physical performance in patients with KOA [5,13,14,15,16,17]. From a specificity perspective, evaluating maximal and explosive knee extensor strength with unilateral tasks makes sense, as several daily activities involve consecutive unilateral actions, such as walking and stair climbing. In addition to these tasks, however, patients with KOA also perform simultaneous bilateral actions, such as squatting and sit-to-stand transfers, quite frequently during the day [18,19,20]. For elderly patients, in particular, knee extensor strength plays an important role in this latter bilateral task, as they need 80% of their maximum knee extensor moment to stand up from a chair [21]. Thus, the use of unilateral tasks (involving separate one-legged contractions) for evaluating knee extensor strength asymmetries in patients with KOA could be questioned against the use of bilateral tasks (involving simultaneous two-legged contractions). In athletes, for example, inter-limb asymmetries in explosive strength—but not in maximal strength—have recently been found to be larger for unilateral than for bilateral tasks [22]. In elderly men and women, Häkkinen et al. [23] demonstrated that both maximal and explosive knee extensor strength did not differ significantly between bilateral and unilateral (sum of the two sides) assessments, however their subjects were healthy, and they did not report inter-limb asymmetries.

The clinical relevance of unilateral vs. bilateral tasks for the assessment of knee extensor strength in patients with KOA remains to be verified. Thus, the primary aim of this cross-sectional study was to compare inter-limb asymmetries of maximal and explosive knee extensor strength between unilateral and bilateral tasks in patients with KOA. A secondary aim was to explore the relationships between unilaterally vs. bilaterally measured strength, performance-based and self-reported function.

## 2. Materials and Methods

### 2.1. Participants

Patients between 50 and 80 years of age with unilateral KOA and scheduled for knee replacement surgery at the Schulthess Clinic (Zurich, Switzerland) were first contacted by telephone. The main exclusion criteria were pain at rest (greater than 3 on a 0–10 scale) in the contralateral knee and any previous open surgery or anterior cruciate ligament reconstruction on either limb. Further exclusion criteria were less than 90° of knee flexion in one of the knees, severe cardiovascular, respiratory, or neurological disorders, open wounds and tissue injuries on the thigh or leg area, and insufficient knowledge of the German language. A convenience sample of 30 patients met the inclusion criteria and took part in the study. Of these, six patients attended only the first of the two scheduled appointments, and therefore 24 patients (12 women and 12 men) completed the entire study procedure.

### 2.2. Study Design

The study was approved by the ethics committee of the Canton of Zurich (Switzerland). Patients gave written informed consent prior to participation. They were first invited to a familiarization session during which the entire test procedure was explained, shown, and practiced. The actual measurements were then carried out at a subsequent appointment (mean interval between familiarization and experimental session: 6 days). The experimental session encompassed the collection of demographic characteristics (i.e., age, sex, weight, and height), the realization of performance-based tests and the evaluation of knee extensor strength. Self-reported function was evaluated with questionnaires completed in the context of the pre-operative medical appointments.

### 2.3. Assessments

#### 2.3.1. Performance-Based Tests

Patients performed a series of four different functional tests according to the recommendations of the Osteoarthritis Research Society International [24]. For these tests, a stopwatch was necessary, usual footwear was worn, and when walking aids or handrail were required, it was documented. In addition, patients also completed a conventional lower limb power test [25]. Before the tests, patients warmed up by cycling for 5 min on an ergometer with a workload of 1 W/kg of body weight.

##### Timed up and Go Test

Patients started the test in a sitting position from a standard chair (seat height: 48 cm; armrest height: 66 cm), with their back leaning and their hands on the armrests. They stood up, walked to a mark 3 m away, turned around, and returned to sit back in the chair at a regular pace (two trials separated by 30 s). The total time (s) until the patients was seated again with his/her back against the back of the chair was recorded. Only the fastest of the two trials was retained.

##### 30-s Chair-Stand Test

Patients started the test in a sitting position from a straight back chair (seat height: 48 cm), with the feet flat on the floor (shoulder-width apart) and arms crossed on the chest. They stood up completely so that hips and knees were fully extended. Then, the patients sat down again so that the buttocks touched the seat completely. This was performed as fast as safely possible, and repeated as often as possible within 30 s (one trial). The total number of complete chair stands (up and down represents one stand) were counted. If at least a full stand was completed at 30 s, this was counted in the total.

##### 40-m Fast-Paced Walk Test

The patients walked as fast but as safely as possible, without running, along a 10-m walkway. Then, they turned around a cone (2 m beyond the start/finish of the walkway) and repeated the walking trial again for a total distance of 40 m (three turns). The total time (s) taken to walk the 4 × 10 m test distance was recorded and expressed as speed (m/s) by dividing the distance (40 m) by the time. The timing was paused during the turns.

##### 9-Step Stair-Climb Test

Patients ascended and descended a flight of stairs with nine steps (step height: 18 cm) and a handrail as quickly as possible but in a safe manner (one trial). The total time (s) to ascend and descend the stairs was recorded.

##### Lower Limb Power

Lower limb power was measured using the Nottingham leg power rig (University of Nottingham, Nottingham, UK). Patients were comfortably seated on the dynamometer chair with the arms crossed in front of the chest, so that their knee angle was ~90°. They were instructed to push the footplate away as quickly as possible by extending the lower limb(s). Measurements were performed three times per task, with rest periods of 30 s between each trial. The tasks were unilateral with the involved side and simultaneous bilateral, presented randomly. During the unilateral task, the free foot was placed on the floor. The final angular velocity of the flywheel was measured by an optoswitch and used to calculate the maximal lower limb power. For each task, the mean of the three trials was retained.

#### 2.3.2. Self-Reported Function

Patients received three different routine questionnaires by e-mail or post before surgery. They completed the questionnaires at home and returned them to the clinic.

##### Oxford Knee Score

Patients filled out the cross-culturally adapted and validated German version of the Oxford Knee Score [26]. The Oxford Knee Score includes 12 questions asking patients to report their knee pain and function over the past 4 weeks. Each item operates with a 5-point response scale with scores ranging from 0 to 4. A total score is calculated by summing up the scores of each item. The total score ranges from 0 to 48, with 0 being the worst possible score and 48 being the best score (indicating very good knee joint function).

##### Core Outcome Measure Index—Knee (COMI-Knee)

The COMI-knee is a valid questionnaire for assessing patients undergoing total knee arthroplasty [27]. The COMI-knee includes six items. With these items, five dimensions are assessed: pain (item 1), function (item 2), symptom-specific well-being (item 3), general quality of life (item 4), and “disability” (average of items 5 and 6). The pain scale is scored from 0 to 10, while the five response options of the items 2 to 6 are scored as 0, 2.5, 5, 7.5, and 10 (for responses 1 to 5, respectively). The average of the five domains gives a COMI score from 0 to 10, with 0 being the best score and 10 being the worst possible score (indicating worst status of the patient). No missing answers are allowed.

##### Euroqol-Five Dimensions (EQ-5D)

The EQ-5D is a valid instrument for measuring health-related quality of life and has been developed for use in a variety of health conditions [28]. The five individual items of the EQ-5D are mobility, self-care, usual activities, pain/comfort, and anxiety/depression. Each item can be rated on a 3-point scale. For the EQ-5D, summary index scores (ranging from −0.59 to 1) were calculated according to the method of Prieto and Sacristán [29], where −0.59 represents the worst state of health and 1 the best.

#### 2.3.3. Maximal and Explosive Strength

Patients were comfortably seated with approximately 65° of knee flexion and approximately 100° of hip flexion on an isometric knee dynamometer (S2P, Ljubljana, Slovenia). The rotational axis of the dynamometer lever arm was aligned with the center of rotation of the knee joint (medial femoral condyle). The distal part of the lever arm was strapped 4–5 cm above the lateral malleolus and each thigh was fixed with a strap in the distal part of the thigh. Additionally, patients were fixed on the chair with a belt across the abdomen to limit movements of the pelvis and their arms were crossed in front of their chest. If a limb was not tested, it was taken out of the strap. The dynamometer had minimal deformability during high-force isometric contractions as it was stable and rigid. For this reason, accurate and reliable evaluation of MVC torque and RTD was possible (intraclass correlation coefficients > 0.97 have previously been reported using the same set-up) [30,31]. On each side, an embedded strain gauge-based force sensor (model Z6FC3-200 kg, Hottinger-Baldwin Messtechnik GmbH, Darmstadt, Germany) assessed the knee joint torque. Force sensors were installed in the frame of the dynamometer so that they operated as torque sensors on a fixed force-sensor lever arm. Torque signals were amplified and analog-to-digital converted (INSAmp, Isotel, Logatec, Slovenia) before being recorded at 1 kHz using a custom-built software (ARS dynamometry, S2P, Ljubljana, Slovenia). The signals were off-line filtered with a smoothing filter (moving average filter) with a window length of 0.03 s.

Patients started each contraction after a verbal signal (3-2-1-go) was given by the examiner. Visual feedback of knee extension strength was consistently shown as real-time torque output of the dynamometer on a large screen. Strong verbal encouragement was consistently provided by the same examiner. Patients were instructed to avoid any countermovement or pre-tension, and additional feedback was provided, if necessary. Torque values were collected during separate unilateral and simultaneous bilateral tasks for both gradual and explosive MVCs of the knee extensors (Figure 1). Gradual MVCs were always performed before the explosive MVCs. The order of tasks (unilateral with the involved side, unilateral with the uninvolved side, and simultaneous bilateral) was randomized within each contraction type. Before strength measurements, patients performed 3–5 submaximal gradual MVCs and 3–5 submaximal explosive MVCs for warm up/familiarization purposes, during which they were asked to reach at least 80% of their maximum estimated strength. For both gradual and explosive MVCs, the interval between the submaximal familiarization trials and the maximal trials was approximately 2 min.

To perform isometric gradual MVCs, patients were instructed to contract their knee extensors “as hard as possible” using a progressive torque build-up. The duration of the build-up phase was quite variable between subjects, but in general it was comprised between 1 and 2 s. The MVC torque should then be held for 2–3 s. Participants were asked to perform two trials, with a rest interval of 30 s in-between. If the MVC torque values of the two trials were more than 10% apart, another trial was carried out and the trial with the lowest MVC torque was excluded. To perform isometric explosive MVCs, patients were instructed to contract their knee extensors “as fast and hard as possible” until they reached the maximum torque and then to relax completely. These contractions lasted approximately 1 s. Participants were asked to perform five trials, with rest intervals of 20 s in-between.

##### Data Analysis

Trials were analyzed under the condition that no pretension and no countermovement (pre-contraction torque changes >2.5% of MVC torque) were observed. Regarding gradual MVCs, the mean value of the 2 MVC torques was used for all subsequent analyses. For each gradual MVC, the MVC torque corresponded to the average torque signal during a 1-s time interval around the maximal torque, which was automatically chosen by the software. Regarding explosive MVCs, the three trials with the highest peak RTD values were further analyzed. The mean value of these trials was used for all subsequent analyses. RTD was calculated as the Δtorque/Δtime value for two time intervals: 0–100 and 0–200 ms relative to torque onset (respectively named RTD 0–100 and RTD 0–200). Torque onset was automatically set by the software as the time point at which the torque signal exceeded a level of 10 Nm. Additionally, peak RTD was quantified as the highest positive value from the first derivative of the torque signal, i.e., the greatest slope of the torque–time curve.

For all strength outcomes (MVC torque, peak RTD, RTD 0–100, and RTD 0–200) and tasks (unilateral and bilateral), inter-limb asymmetries between the uninvolved and involved side were calculated according to this equation [12]:$$\mathrm{Inter}\text{-}\mathrm{limb}\;\mathrm{asymmetry}\;\left(\%\right)=\frac{\mathrm{uninvolved}-\mathrm{involved}}{\mathrm{uninvolved}}\times 100$$

To better compare knee extensor strength and lower limb power between patients, absolute strength and power were also normalized as a function of body weight, to provide relative strength/power data.

### 2.4. Statistical Analysis

Statistical analyses were performed using Jamovi software version 1.0.4.0 (https://www.jamovi.org/ (accessed on 27 July 2021)). Data normality was verified with Shapiro–Wilk tests. Normally distributed data were expressed as means and standard deviations (SD) and non-normally distributed data were expressed as medians and interquartile range. First, absolute data for all strength outcomes (MVC torque, peak RTD, RTD 0–100, and RTD 0–200) were evaluated with a two-way repeated measures ANOVA (side (involved and uninvolved) and task (unilateral and bilateral)). Inter-limb asymmetries were also evaluated with a two-way repeated measures ANOVA (task (unilateral, bilateral) and strength outcome (MVC torque, peak RTD, RTD 0–100, and RTD 0–200)). If the assumption of sphericity was violated, the Greenhouse–Geisser correction was applied. For both ANOVAs, Tukey post hoc tests were performed in case of significant main effect or interaction. The relationships between relative strength and performance-based function or self-reported function were assessed with Pearson’s correlation coefficient or Spearman’s rank coefficient tests, respectively. The level of significance was set at *p* ≤ 0.05 for all analyses.

## 3. Results

Patient characteristics, as well as performance-based and self-reported function results, are presented in Table 1. Absolute strength data are reported in Table 2. For all strength outcomes and tasks (unilateral and bilateral), the involved side showed significantly lower MVC torque and RTD values compared to the uninvolved side (*p* < 0.001). Similarly, bilaterally measured strength outcomes were consistently lower than unilaterally measured outcomes (*p* < 0.05). Inter-limb asymmetries are reported in Figure 2. For each strength outcome, inter-limb asymmetries did not differ significantly between unilateral and bilateral tasks (*p* > 0.05), and additionally no differences were observed between the different strength outcomes (*p* > 0.05). Unilateral vs. bilateral mean (SD) inter-limb asymmetries were, respectively, 22% (15) vs. 19% (16) for MVC torque, 25% (17) vs. 23% (25) for peak RTD, 28% (16) vs. 23% (24) for RTD 0–100, and 28% (12) vs. 23% (21) for RTD 0–200.

Relative strength outcomes correlated significantly with all performance-based tests (*p* < 0.05), except the 30-s chair-stand test (Table 3), with comparable correlation coefficients between unilateral and bilateral tasks. Representative correlations between unilaterally and bilaterally measured RTD 0–100 and the 40-m fast-paced walk test are presented in Figure 3. There was no significant correlation between any unilateral or bilateral relative strength outcomes and self-reported function (*p* > 0.05) (Table 3).

## 4. Discussion

The main findings of this study were that inter-limb asymmetries of maximal and explosive knee extensor strength did not differ significantly between unilateral and bilateral tasks in patients with KOA. In the same way, the relationships between knee extensor strength—measured either unilaterally or bilaterally—and performance-based/self-reported function were not influenced by the type of task. Thus, unilateral and bilateral tasks have an equivalent clinical relevance for the assessment of maximal and explosive knee extensor strength in patients with KOA.

### 4.1. Absolute Strength

For all strength outcomes and tasks, isometric knee extensor strength was lower on the involved side compared to the uninvolved side. This result is consistent with the meta-analysis of Moon et al. [7] for maximal strength (MVC torque), and also with the studies of Callahan et al. [6] and Ventura et al. [8] for explosive strength (different RTD variables). Taken as a whole, these findings confirm the presence of consistent and considerable knee extensor strength deficits (i.e., muscle weakness) in patients with KOA, irrespective of the strength outcome and task. Another consistent finding from our absolute strength comparisons is that bilaterally measured strength was systematically lower than the equivalent unilateral condition. This confirms the presence of the so-called bilateral deficit [32] for all maximal and explosive strength outcomes in patients with KOA.

### 4.2. Inter-Limb Asymmetries

For all strength outcomes, no significant difference was found between inter-limb asymmetries obtained from unilateral and bilateral tasks. Thus, for the quantification of muscle weakness in patients with KOA using side-to-side comparisons, it does not seem to make a difference whether strength is measured unilaterally or bilaterally. This is consistent with the recent findings of Sarabon et al. [22], who also found comparable unilateral and bilateral asymmetries for maximal strength in high-level athletes from different sports. However, they observed larger asymmetries of explosive strength for unilateral than bilateral tasks, which is contrary to the results of the present study. A possible explanation is that most of the athletes they tested are used to performing explosive unilateral actions, such as accelerating, braking, changing direction, jumping, landing, or kicking [33], while these actions are rarely or never performed by KOA patients.

In order to compare our unilateral and bilateral inter-limb asymmetries with similar data reported in the literature, we recalculated asymmetries from previous studies conducted with patients with a unilateral complaints as well as with healthy people (including athletes). The first category, including post-stroke patients [34,35] and patients with anterior cruciate ligament reconstruction [36,37,38,39], showed average unilateral and bilateral inter-limb asymmetries of 21% (SD = 21) and 19% (SD = 18), respectively. In studies with healthy subjects [36,37,38,40,41,42,43,44,45,46,47,48,49,50,51,52,53,54,55], unilateral and bilateral inter-limb asymmetries were, on average, 7% (SD = 9) and 8% (SD = 9), respectively. These data taken as a whole seem to confirm the lack of difference between unilateral and bilateral inter-limb asymmetries, in line with our main findings. However, it must be mentioned that the studies cited above used a variety of experimental conditions (different tests, tasks, joints, etc.), outcomes, and patients/subjects (for which the mechanisms underlying muscle weakness are probably not the same), and in some cases different unilateral and bilateral tasks were used within the same study [36,39,50]. Therefore, these analyses must be interpreted with caution. Contrary to our current study, however, none of these studies evaluated the relationship between unilaterally vs. bilaterally measured strength outcomes and performance-based or self-reported function.

### 4.3. Performance-Based Function

Overall, the present study disclosed significant correlations between relative knee extensor strength, either measured unilaterally or bilaterally, and performance-based tests, except for the 30-s chair-stand test. These results are in agreement with those from studies examining patients scheduled for total knee arthroplasty, which also showed an association between knee extensor strength and performance-based tests [17,56,57]. A possible explanation for the lack of correlation between knee extensor strength and the 30-s chair-stand test is the relatively high number of repetitions completed by our patients (15.6), which is higher than the 10–11 repetitions reported by Brown et al. [56] and Skoffer et al. [16]. In fact, as the number of repetitions increases, strength endurance may contribute more to the overall sit-to-stand performance than pure knee extensor strength, and this could have contributed to reduce the degree of correlation between the 30-s chair-stand test and maximal/explosive strength in our study.

Surprisingly, performance of daily activities entailing unilateral alternated contractions of lower limb muscles (such as walking and stair climbing) did not show higher correlations with unilaterally than bilaterally measured knee extensor strength, and vice versa for daily activities entailing bilateral concomitant contractions (such as sit-to-stand). This result is in line with a study examining patients with KOA, in which similar correlations between unilaterally vs. bilaterally measured one repetition maximum strength and performance-based tests were observed [58]. Therefore, when considering limitations in activities of daily living in patients with KOA, it does not seem to make a difference whether knee extensor strength is assessed with a unilateral or bilateral task.

### 4.4. Self-Reported Function

There was no significant correlation between relative knee extensor strength and self-reported function, and it was equally true for both unilateral and bilateral tasks. This finding is consistent with earlier studies from Skoffer et al. [16] and Tevald et al. [58], who also found no relationship between the performance of unilateral or bilateral tasks and self-reported function in individuals with KOA. An explanation for the lack of correlation between involved-side strength and patient-reported outcomes, such as knee pain and function, as well as health-related quality of life, could be the non-negligible contribution of the uninvolved side to self-reported function. In fact, a recent study unsurprisingly reported significant correlations between uninvolved-side knee extensor and flexor strength and self-reported function in individuals with KOA [59], which partially support this theory.

### 4.5. Implications

In this study, we examined the clinical relevance of unilateral and bilateral tasks for the evaluation of knee extensor strength by comparing the magnitude of inter-limb asymmetries and the degree of correlation with physical function between unilaterally and bilaterally measured maximal and explosive strength. As both asymmetry and correlation results did not reveal an advantage for one of the two tasks, we suggest that both unilateral and bilateral tasks can be used with confidence—though not interchangeably—to evaluate maximal and explosive knee extensor strength in patients with KOA. However, there are other considerations for clinical use that should be addressed. On the one hand, it is important to mention that bilateral measurements require fewer trials (actually half compared to unilateral tests) and thus less burden for both patients and clinical staff. On the other hand, bilateral assessments are only possible with double-sensor dynamometers, such as the one used in this study. These dynamometers are however less common in clinical practice—though not necessarily more expensive than single-sensor tools—as knee extensor strength is predominantly measured with unilateral tasks. Another important clinical relevance criterion for the choice of unilateral vs. bilateral tasks is the level of pain/discomfort perceived by patients during the maximal contraction, which could potentially be lower during bilateral tasks. However, we observed no difference in the amount of knee joint pain during both gradual and explosive contractions between unilateral (mean pain: 0.7 (SD = 0.9)) and bilateral (mean pain: 0.6 (SD = 0.9)) tasks, as assessed with a visual analogue scale (0–10 scale) immediately after each contraction. In the same way, this pain measure did not correlate significantly with inter-limb asymmetries, nor with absolute strength values in all the conditions, which suggests that knee pain is not the main determinant, but rather one of the possible contributors to knee extensor muscle weakness in patients with KOA. Finally, whether unilateral and bilateral tasks/exercises may have the same treatment effectiveness within a standardized rehabilitation/strength training protocol is highly speculative, but certainly worth investigating.

### 4.6. Study Limitations

This study has several limitations that should be acknowledged. First, our findings cannot be generalized to all individuals with KOA, as our patients were recruited in a private hospital and probably had a better function and health state compared to the general population of total knee replacement candidates. This is evident, for example, from the Oxford Knee Score, which was clearly higher in the present study (28 points) compared to other investigations (ranging between 18 and 20 points [60,61,62]; the minimal clinical important difference for the Oxford Knee Score being 3 to 5 points [63]), but also from the performance-based test results (already discussed). Second, we only considered two criteria to examine the clinical relevance of unilateral and bilateral tasks (inter-limb asymmetries and relationships with function), but we did not quantify, for example, the respective inter-session and inter-observer reliability of unilateral and bilateral tasks. Third, inter-limb asymmetries were calculated by assuming that one of the two knees was “uninvolved” (i.e., the one not scheduled for knee replacement), which is certainly a simplistic classification for KOA patients who very often present with bilateral signs and symptoms. This could have contributed to the between-subject variability in inter-limb asymmetry data, and probably influenced some of the analyses. Lastly, due to the relatively small sample size, no subgroup analysis could be conducted regarding sex or KOA severity.

## 5. Conclusions

To conclude, it does not seem to make a difference in terms of clinical relevance whether maximal and explosive knee extensor strength are evaluated with unilateral or bilateral tasks in patients with KOA.

## Figures and Tables

**Figure 1 jcm-10-04353-f001:**
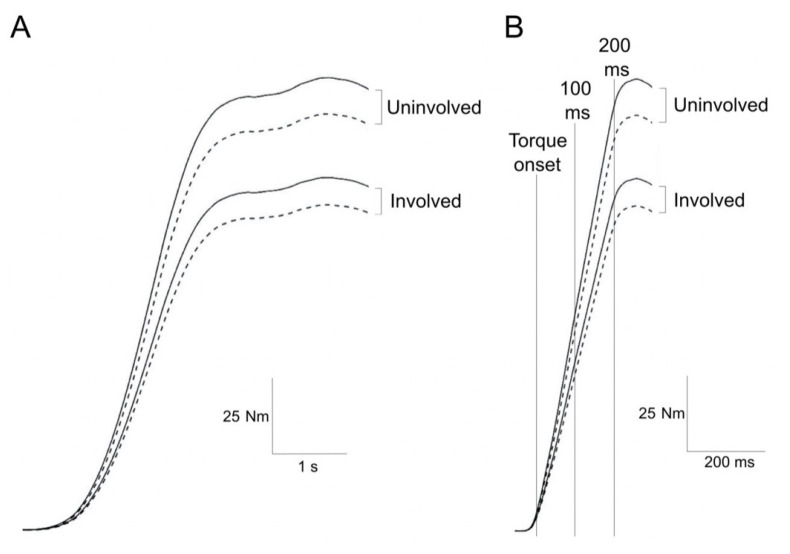
Representative knee extensor torque-time traces by side and task. Unilaterally measured (continuous line) and bilaterally measured (dashed line) knee extension torque recorded during gradual (**A**) and explosive (**B**) maximal voluntary contractions. In (**B**), torque onset and time points of 100 and 200 ms from torque onset are shown as vertical lines.

**Figure 2 jcm-10-04353-f002:**
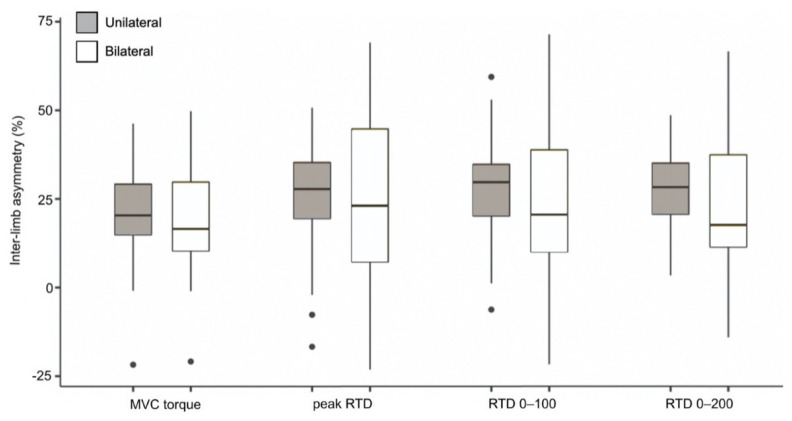
Inter-limb asymmetries by task and strength outcome. The box represents the interquartile range; the horizontal line represents the median. The upper whisker extends from the 75th percentile to the largest value that is no further than 1.5 times the interquartile range from the 75th percentile. The lower whisker extends from the 25th percentile to the smallest value that is no further than 1.5 times the interquartile range from the 25th percentile (outliers are also shown).

**Figure 3 jcm-10-04353-f003:**
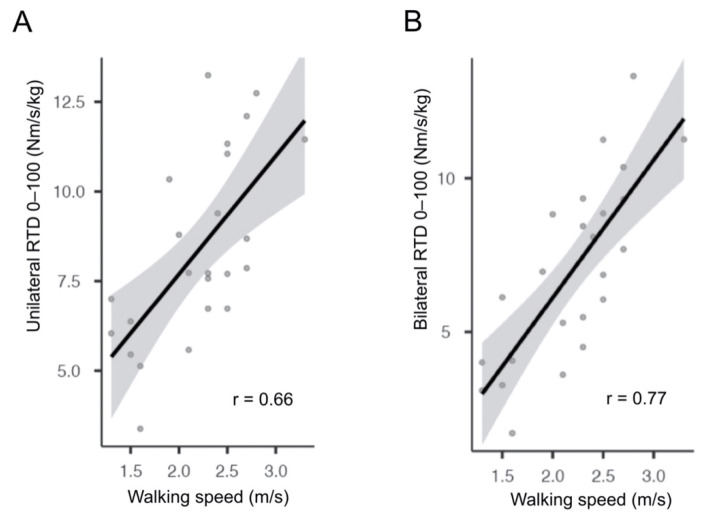
Representative correlations between explosive strength and performance-based function. Correlation plots with standard errors (shaded area) and correlation coefficients (r) for unilateral (**A**) and bilateral (**B**) relative RTD 0–100 (involved side) against 40-m fast-paced walking speed.

**Table 1 jcm-10-04353-t001:** Patient characteristics, and performance-based and self-reported function results.

Variable	Mean (SD)
Age (yrs)	65 (7)
Weight (kg)	87 (26)
Height (cm)	170 (10)
Timed up and go test (s)	5.97 (1.43)
30-s chair-stand test (repetitions)	15.60 (3.45)
40-m fast-paced walk test (m/s)	2.20 (0.52)
9-step stair-climb test (s)	10.10 (4.32)
Relative lower limb power-involved (W/kg)	1.59 (0.69)
Relative lower limb power-bilateral (W/kg)	2.83 (0.76)
	Median (IQR)
Oxford Knee Score (0 = worst; 48 = best)	28.0 (25.5–32.8)
COMI-knee (0 = best; 10 = worst)	6.05 (5.10–6.56)
EQ-5D index (−0.59 = worst; 1 = best)	0.78 (0.65–0.83)

COMI, Core Outcome Measure Index; EQ-5D, Euroqol-Five Dimensions; and IQR, interquartile range.

**Table 2 jcm-10-04353-t002:** Absolute strength outcomes by side and task.

	Involved Side		Uninvolved Side	
	Unilateral	Bilateral	Unilateral	Bilateral
MVC torque (Nm)	149 (49) °	141 (51) ° *	191 (59)	176 (61) *
Peak RTD (Nm/s)	843 (390) °	709 (943) ° *	1129 (488)	943 (485) *
RTD 0–100 (Nm/s)	718 (299) °	607 (306) ° *	1011 (400)	835 (434) *
RTD 0–200 (Nm/s)	549 (213) °	480 (227) ° *	761 (282)	633 (295) *

Values are presented as mean (SD). MVC, maximal voluntary contraction; RTD, rate of torque development. °, Involved < uninvolved at *p* ≤ 0.001 (Tukey post hoc); and *****, Bilateral < unilateral at *p* ≤ 0.05 (Tukey post hoc).

**Table 3 jcm-10-04353-t003:** Correlations between unilateral and bilateral relative strength, performance-based and self-reported function.

	MVC Torque		Peak RTD		RTD 0–100		RTD 0–200	
	Unilateral	Bilateral	Unilateral	Bilateral	Unilateral	Bilateral	Unilateral	Bilateral
Performance-based function								
Timed up and go test	–0.70 ***	–0.73 ***	–0.58 **	–0.63 ***	–0.53 **	–0.65 ***	–0.60 **	–0.69 ***
30-s chair-stand test	0.31	0.35	–0.01	0.00	0.00	0.02	0.12	0.04
40-m fast-paced walk test	0.62 **	0.64 ***	0.69 ***	0.71 ***	0.66 ***	0.77 ***	0.58 **	0.74 ***
9-step stair-climb test	–0.61 **	–0.67 ***	–0.59 **	–0.55 **	–0.56 **	–0.55 **	–0.57 **	–0.60 **
Relative lower limb power-involved	0.66 ***	0.60 **	0.66 ***	0.61 **	0.66 ***	0.71 ***	0.58 **	0.62 **
Relative lower limb power-bilateral	0.52 **	0.50 *	0.56 **	0.55 **	0.49 *	0.56 **	0.49 *	0.51 *
Self-reported function								
Oxford Knee Score	0.14	0.17	0.14	–0.05	0.17	–0.04	0.17	–0.07
COMI-knee	0.00	–0.01	0.06	0.15	0.04	0.18	0.04	0.18
EQ-5D index	–0.07	–0.07	–0.12	–0.25	–0.01	–0.25	–0.07	–0.15

COMI, Core Outcome Measure Index; EQ-5D, Euroqol-Five Dimensions; MVC, maximal voluntary contraction; and RTD, rate of torque development. For performance-based function, values represent Pearson’s correlation coefficients. For self-reported function, values represent Spearman’s rank correlation coefficients. * *p* < 0.05, ** *p* < 0.01, and *** *p* < 0.001.

## Data Availability

The data that support the findings of this study are available from the corresponding author upon reasonable request.

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
