# Peer review of "Should We Use Unilateral or Bilateral Tasks to Assess Maximal and Explosive Knee Extensor Strength in Patients with Knee Osteoarthritis? A Cross-Sectional Study"

_jcm, 2021, doi:10.3390/jcm10194353_

Round 1
Reviewer 1 Report
Please see attached file for reviewer comments and suggestions.

Reviewer 2 Report
An important topic is discussed in the paper.
As the population ages, the number of people suffering from osteoarthritis of the knee increases. There is a need for an objective assessment of the functional status of these patients. The work contains an appropriate introduction, the methods have been selected and described correctly, and appropriate statistical tests have been applied. The work was written interestingly. The authors are aware of the limitations and describe them.
I have only one comment. Please check line 247. Why did you use a significance level p> 0.25 in the Shapiro-Wilk test (the usual p> 0.05 is used)? Is it a mistake?
Reviewer 3 Report
The study entitled „Should we use unilateral or bilateral tasks to assess maximal and explosive knee extensor strength in patients with knee osteoarthritis? A cross-sectional study.“ by Pfeifle and colleagues was conducted with the aim of analyzing the clinical relevance of unilateral vs. bilateral tasks for assessing knee extensor strength in patients with knee osteoarthritis. The study is well designed and the materials and methods are adequately described. The results are understandably presented textually, tabularly and graphically, and thoroughly analyzed and compared with other research in the discussion.
Below, please find suggestions for minor revision:
Lines 34-36: With an increasing number of knee osteoarthritis cases worldwide, you should use the most recent data on KOA prevalence, including the most recent references.
Some references are not written according to the journal's instructions. Please consult the instructions on the link: https://www.mdpi.com/journal/jcm/instructions#referees
